# Sepsis among Neonates in a Ghanaian Tertiary Military Hospital: Culture Results and Turnaround Times

**DOI:** 10.3390/ijerph191811659

**Published:** 2022-09-16

**Authors:** Francis Kwame Morgan Tetteh, Raymond Fatchu, Kingsley Ackah, Trudy Janice Philips, Hemant Deepak Shewade, Ama Pokuaa Fenny, Collins Timire, Jeffrey Karl Edwards, Emmanuel Abbeyquaye Parbie

**Affiliations:** 1Pathology Division, 37 Military Hospital, Neghelli Barracks, Accra GA 01, Ghana; 2Clinical Pathology Department, Noguchi Memorial Institute for Medical Research, P.O. Box LG 581, Accra GA 01, Ghana; 3Division of Health System Research, ICMR-National Institute of Epidemiology (ICMR-NIE), Chennai 600077, India; 4Institute of Statistical, Social and Economic Research, University of Ghana, P.O. Box LG 1181, Accra GA 01, Ghana; 5International Union against Tuberculosis and Lung Disease (The Union), 75006 Paris, France; 6Department of Global Health, University of Washington, Seattle, WA 98195, USA; 7Paedriatric Division, 37 Military Hospital, Neghelli Barracks, Accra GA 01, Ghana

**Keywords:** neonatal sepsis, bacteria, neonatal intensive care unit, turnaround time, antibiotic resistance, sort it, operational research

## Abstract

In this study, we described the bacterial profile, antibiotic resistance pattern, and laboratory result turnaround time (TAT) in neonates with suspected sepsis from a tertiary-level, military hospital in Accra, Ghana (2017–2020). This was a cross-sectional study using secondary data from electronic medical records. Of 471 neonates clinically diagnosed with suspected sepsis in whom blood samples were collected, the median TAT from culture request to report was three days for neonates who were culture-positive and five days for neonates who were culture-negative. There were 241 (51%) neonates discharged before the receipt of culture reports, and of them, 37 (15%) were culture-positive. Of 471 neonates, twenty-nine percent (*n* = 139) were bacteriologically confirmed, of whom 61% (*n* = 85) had late-onset sepsis. Gram-positive bacterial infection (89%, *n* = 124) was the most common cause of culture-positive neonatal sepsis. The most frequent Gram-positive pathogen was coagulase-negative *Staphylococcus* (55%, *n* = 68) followed by *Staphylococcus aureus* (36%, *n* = 45), of which one in two were multidrug resistant. The reasons for large numbers being discharged before the receipt of culture reports need to be further explored. There is a need for improved infection prevention and control, along with ongoing local antimicrobial resistance surveillance and antibiotic stewardship to guide future empirical treatment.

## 1. Introduction

Neonatal sepsis remains the most common cause of neonatal morbidity and mortality worldwide, particularly in low- and middle-income countries [1]. Neonatal sepsis is defined as an infection with systemic signs which is diagnosed initially clinically, then confirmed through blood culture, within the first four weeks of life [2]. Neonatal sepsis is sub-classified by the time of presentation into early onset sepsis (EOS) if within 72 h of birth, and late-onset sepsis (LOS) if it occurs from three to 28 days of age [3]. EOS is typically acquired by exposure to microorganisms that colonize the maternal genital tract (vertical transmission) during the perinatal period [4]. Group B *Streptococcus* is the most common isolated bacteria in developed countries, though other bacteria have been implicated [4,5]. LOS is usually caused by pathogens from the home or hospital environment [4,5]. *Staphylococcus* species and Gram-negative bacteria, mostly from the *Enterobacteriaceae* family, are the most frequently identified pathogens with LOS [5,6].

Even though neonates are particularly susceptible to sepsis and subsequent high mortality, few countries provide incidence data for this age group [7]. The 2016–2017 global burden of disease study estimated an incidence of 1.3 million cases of neonatal sepsis [7,8]. During 1979–2019, neonatal sepsis incidence was estimated at 2824 per 100,000 live births, of which an estimated 17.6% died [6]. The estimated incidence and mortality were higher in EOS than in LOS. From 2009 to 2018, the incidence of both EOS and LOS in low- and middle-income countries (3930 per 100,000 live births) was higher than the global incidence [6]. Within Africa, the neonatal mortality ratio is estimated at approximately 27 per 1000 live births [9]. Sub-Saharan Africa accounts for 36% of neonatal deaths worldwide [10]. The projected economic cost associated with neonatal sepsis and mortality in this region is up to USD 469 billion (2014 data) [11].

The high rates of neonatal sepsis-related mortality and the types of pathogens commonly implicated in developing countries suggest that the lack of appropriate hygiene during labor and delivery, postnatal care, and feeding are the key contributors [4]. To reduce this burden, improved infection-control practices (i.e., hand hygiene, isolation, aseptic techniques, disinfection, or sterilization of equipment) have been recommended [4].

Neonatal sepsis is initially treated empirically with broad-spectrum antibiotics while waiting for culture confirmation [1]. This approach likely increases the risk of antimicrobial-resistant (AMR) bacteria, leading to an increased risk of mortality for neonatal sepsis [1,12]. Though empirical antibiotic treatment is used, it should be ideally guided by local susceptibility patterns and the Access, Watch and Reserve (AWaRe) grouping of antibiotics developed by the World Health Organization (WHO), to minimize the possibility of AMR [13]. Knowledge of the local antibiotic resistance profiles has been demonstrated to improve the probability of selecting effective empirical therapy in low-resource settings, which should lead to lower AMR risk [14].

In Ghana, where a neonate dies approximately every 15 min, neonatal mortality is an enormous problem [15]. The Greater Accra region, where the capital city of Ghana is located, has an estimated neonatal mortality ratio of 25 deaths per 1000 live births [16]. There is a paucity of reports on neonatal sepsis from the West African sub-region and Ghana specifically. Findings of significant antibiotic resistance, including multidrug resistance (MDR), have been previously reported in two studies from Ghana, one conducted in 2010–2013 at a teaching hospital [5], and the other in 2016 at two public hospitals [17]. The culture positivity rate for suspected neonatal sepsis was approximately 20% in both studies. However, neither study measured the turnaround time (TAT) from collection of the blood sample to culture and antibiotic susceptibility testing (AST) reporting, which is important in choosing the most effective antibiotics to reduce morbidity, mortality, and AMR risk. Additionally, further study is needed to evaluate if local pathogens and antibiotic susceptibility patterns are changing over time.

Therefore, in a large tertiary-level military hospital in Accra, Ghana (2017–2020), we aimed to determine the culture positivity, the TAT, and the availability of the culture report before discharge among neonates with suspected sepsis. We then described the bacterial profile and antibiotic susceptibility pattern (including MDR) among neonates with confirmed (culture-positive) sepsis.

## 2. Materials and Methods

### 2.1. Study Design

This was a hospital-based cross-sectional study using secondary data.

### 2.2. Setting

#### 2.2.1. General Setting

Ghana is a country located in West Africa along the Gulf of Guinea and the Atlantic Ocean [18] and has a population of approximately 32 million [19]. Healthcare in Ghana is largely administered by the Ministry of Health and the Ghana Health Service [20]. Ghana has instituted a national universal medical insurance system, the National Health Insurance Scheme (NHIS), which covers most basic inpatient and primary care outpatient services for patients that purchase the insurance [20]. However, those who do not enroll in the NHIS are left to pay for all services and medications out of pocket.

#### 2.2.2. Specific Setting

The “37 Military Hospital” is a 600-bed level 4 hospital located in the Ayawaso East Municipality of Accra. It serves mainly the military and their dependents as well as civilians mostly from Accra. The hospital records about 95 admissions per day. The outpatient department records approximately 30,000 patients per month. The hospital covers all medical specialties and provides referral healthcare services to an estimated population of 30 million. For the serving and ex-military personnel, civilian employees, and their < 18-year dependents (called “entitled”), all the services are provided within the hospital free of cost. For civilians not employed by the military and non-dependents (called “non-entitled”), the services are either covered by the NHIS or paid out of pocket. The hospital record-keeping system utilizes both paper-based and electronic medical records (EMR). EMR includes clinical, laboratory, imaging, and pharmacy data.

The neonatal intensive care unit (NICU) has a bed capacity of 29 beds and admits about 60 neonates a month. The nursing staff-to-patient ratio is approximately 1:4. All blood samples from neonates with suspected sepsis are sent to the microbiology laboratory (on-site) for culture and AST after the laboratory request is made electronically. Aside from the NICU, other wards that record neonatal visits but less often include the pediatric emergency unit, pediatric outpatient department, and Nkrumah ward. The empirical antibiotics used for suspected neonatal sepsis are amikacin and ciprofloxacin, according to local and NICU guidelines.

#### 2.2.3. Bacteriological Procedures for Suspected Neonatal Sepsis

For patients with suspected neonatal sepsis, 1–3 mL of blood is inoculated directly into Paediatric Bactec^®^ blood culture vials and incubated for five days in the Bactec Fx blood culture system (Becton Dickinson, NJ, USA) as per the manufacturer’s instructions. Where bacterial growth is detected (on any day) within the five days, initial Gram stains are performed and preliminary results are shared with the attending clinicians to guide their choice of empirical treatment. Subcultures are made onto blood, MacConkey, and Saboraund agars and incubated aerobically at 37 °C for 18 to 24 h. Similarly, the sample is subcultured onto chocolate agar plates and incubated anaerobically at 37 °C for 18 to 24 h. Bacterial isolates are identified using Gram stain and routine biochemical methods. Bacteria speciation and AST are performed using the Pheonix100 identification system (Becton Dickinson, NJ, USA) in accordance with the clinical and laboratory standards institute guidelines [21]. Resistance to oxacillin in *S. aureus* isolates is interpreted as methicillin-resistant *S. aureus* [21]. When no bacterial growth is detected after five days of incubation, a final negative culture report is entered electronically onto the EMR and is accessed by the attending clinicians.

### 2.3. Study Population

All admitted neonates with suspected sepsis, whose blood samples were received for culture at the microbiology laboratory from January 2017 through December 2020, formed the study population.

### 2.4. Variables, Sources of Data, and Data Collection

The following variables were extracted from the EMR into Excel 2010 (Microsoft, Redmond, WA, USA): laboratory number (unique identifier), hospital folder number, date of admission, sex, age in days at admission, birth weight in kilograms, ward (NICU, pediatric emergency), type of beneficiary (entitled/non-entitled), neonatal sepsis categorization (EOS/LOS), date of culture request, outcome (discharged after receiving culture report, discharged before receiving culture report death), date of outcome, date of culture report along with AST, culture result (non-contaminant growth, contaminant, no growth), isolate (species and Gram +/−), sensitivity to antibiotics (sensitive or resistant).

For positive cultures, organisms including *Micrococcus* spp., *Bacillus* spp., and Diphtheroids were classified as contaminants. An isolate was considered to be multidrug resistant when resistance was observed for at least one agent in three or more antimicrobial categories [22]. TAT (in days) was calculated using the date of culture request and the date of culture report (available in the EMR). The proportion of culture reports released after discharge or death was inferred based on the date of outcome and the date of culture report.

The source of data was from the EMR and the variables extracted were cross-checked record by record, using the paper-based database in the laboratory and the wards (NICU/pediatric emergency/others). Duplicates were removed using the hospital folder number. If a child had more than one culture, the first one was considered for the purpose of this study. During the data extraction process, we de-identified blood culture reports to ensure complete anonymity from laboratory archives.

### 2.5. Data Analysis

Patient-level data were cleaned and imported to EpiData analysis software (version 2.2.2.186, EpiData Association, Odense, Denmark). Numbers and proportions were used to summarize categorical variables. TAT and admission duration (in days) were presented as median and interquartile ranges (IQR). Antibiotic resistance, stratified across the AWaRe group of antibiotics and derived as MDR, was also described. Differences in proportions were assessed for statistical significance using chi-square and chi-square for trend test, as appropriate.

## 3. Results

### 3.1. Characteristics of Neonates with Suspected Sepsis

There were 471 blood samples collected from neonates with suspected sepsis (Figure 1 and Table 1). Over the four-year period, the largest number of suspected cases with blood samples collected were reported in 2019 (220, 47%). The majority of neonates were <7 days of age (72%), of normal birth weight (69%), and from the non-entitled group (83%) (72%).

### 3.2. Turnaround Time and Admission Outcomes of Neonates with Suspected Sepsis

The median TAT from sample submission to culture and AST reports was five days (IQR 3–5): it was three days for culture-positive samples and five days for culture-negative samples. Of the 471 neonates, 439 (93%) clinically improved and were discharged, while 32 (7%) died. Of 471 neonates, 241 (51%) received the culture report after the admission outcome and of them, 37 (15%) were culture positive. Of the 32 neonates who died, 19 received the culture report after death and 11% of those had positive cultures. The median admission duration was four days (IQR 4–6), six days for those that were discharged after receiving culture results, and three days for those who either died or were discharged before receiving culture results.

### 3.3. Characteristics of Neonates with Culture Confirmed Sepsis

Of the 471 neonates with suspected sepsis, 139 (29%) were confirmed by culture. Of 139 culture-confirmed neonates, 54 (39%) were EOS and 85 (61%) were LOS (Table 2). There was a decreasing trend in the proportion of LOS over the four years, with the highest being 68% in 2017 and the lowest being 54% in 2020, although these differences were not statistically significant (*p* = 0.246). Very low birth weight neonates were more likely to have EOS when compared to low and/or normal birth weight (77% vs. 32%, *p* < 0.001). A similar proportion of entitled/non-entitled neonates had EOS (37% vs. 39%) and LOS (63% vs. 61%). Neonates with EOS and LOS were both more likely to have a Gram-positive etiology for sepsis.

### 3.4. Pathogens Isolated

Among Gram-positive pathogens, coagulase-negative *Staphylococcus* (CoNS) was the most common isolate (55%, 68/124), followed by *Staphylococcus aureus* (36%, 45/124). For Gram-negative infections, *Klebsiella pneumoniae* was the most common isolate (40%, 6/15). CoNS and *S. aureus* were the most frequent pathogens for both EOS (48% and 30%, respectively), and LOS (49% and 34%, respectively).

### 3.5. Antimicrobial Susceptibility including MDR

All culture-confirmed infections were resistant to at least one antibiotic (Table 3 and Table 4). Of the 139 positive cultures, MDR was seen in 71 (51%) isolates. The year wise trend in MDR was: 2017—54%, 2018—58%, 2019—60% and 2020—32% (*p* = 0.141). Gram-positive bacteria were more frequently MDR (67/124, 54%) compared to Gram-negative bacteria (4/15, 27%); however, the difference was statistically non-significant (*p* = 0.121) (Table 5). MDR among EOS was 50% (27/54) and among LOS was 52% (44/85).

The proportion of CoNS and *S. aureus* isolates that were MDR were 52% and 51%, respectively. Gram-positive bacteria showed high resistance to multiple antibiotics including ampicillin (79% CoNS, 89% *S. aureus*), 2nd–3rd generation cephalosporins (57%, CoNS and 29–61% *S. aureus*), methicillin (59% *S. aureus*) and vancomycin (44% CoNS, 76% *S. aureus*). We found a varying range of resistance by Access (0–85%) and Watch (0–63%) categorization (Table 6).

## 4. Discussion

### 4.1. Key Findings

In this study of neonatal sepsis at a large tertiary-level military hospital in Accra, Ghana, during 2017–2020, we had the following key findings: (1) the turnaround times from culture request to report were satisfactory but due to a large number of neonates being discharged within three days, the culture report was not received before discharge for every second neonate; (2) one in three neonates with suspected sepsis were culture-confirmed and three in five neonates with confirmed sepsis were late-onset; (3) the most common pathogen was CoNS, followed by *S. aureus*, of which one in two of the isolates were MDR.

### 4.2. Strengths and Limitations

The strengths of this study include having access to all submitted blood samples for all neonates with suspected sepsis at a single hospital over a four-year period, which limits the chance of bias. This study also included all patients, regardless of the payer source. The major limitation was the use of retrospective data, which limits the collection of more patient and provider-specific level data for analysis. As we used laboratory data from EMR and not the clinical data, we were not able to assess the proportion of neonates with suspected sepsis who did not undergo culture and AST. In addition, this study was from a single capital city hospital setting. The results may be more representative of neonatal care within Ghana as a whole, if multiple facilities were included, particularly from outside of Accra. Finally, study isolates were not checked for resistance to WHO Reserve category antibiotics.

### 4.3. Implications and Recommendations

The median TAT from request for culture to culture results, overall (five days) and stratified by culture confirmation (three days for culture-positive and five days for culture-negative), was expected and not surprising considering the bacteriological procedures used. These results indicate that around half of the culture-confirmed neonates with sepsis may have to wait for three days before a more specific and narrow-spectrum antibiotic choice can be made. Others have reported that the average TAT for blood culture and AST reporting in sepsis should be closer to 3 days and with more advanced technological techniques, could be reduced to 50 h [23]. This “need for speed” could be directly translated into improved patient outcomes and reduced risk of AMR in the future not for only neonates, but for all patients under investigation for sepsis [24]. This is an area where further improvements could be leveraged with a possible high return on investment, especially in low- and middle-income countries where rapid diagnostics remain cost-prohibitive.

Our finding is similar to a study performed in Nepal where the median TAT was approximately six days [25], and where, in around two-thirds of the neonates, the antibiotic was changed based on the culture report. In our study due to a large number of neonates being discharged within three days (we speculate that this was due to an improvement in the clinical condition of the neonate), the culture report was not received before discharge for every second neonate. This needs further in-depth qualitative exploration.

In the present study, clinical sepsis was confirmed (29%) more frequently than in previous reports from Ghana (17% and 22%) [5,16], Saudi Arabia (16%) [26], but was similar to Nepal (29%) [25] and lower than in Myanmar (42%) [27]. Additionally, our proportion of reported culture-confirmed neonates was higher than reported from high-income countries [28]. Low culture confirmation rates of clinical neonatal sepsis have been attributed to multiple etiologies including maternal antibiotics before delivery, antibiotic initiation before culture, difficulty in neonatal sampling, and the challenge of making the clinical diagnosis of sepsis [28]. Our findings suggest that at this hospital, there is improving diagnostic acumen and antibiotic stewardship compared to previous studies from Ghana [5,16].

However, we found that three in five neonates with confirmed sepsis were LOS, which implies that there continue to be challenges in preventing nosocomial infections. This finding supports the need for improved infection control as a primary means of decreasing the risk of LOS.

In the present study, CoNS predominated in culture-confirmed cases (55%) and this has been found to be the major etiology of LOS in neonates [29]. This finding is similar to the most recent study from Saudi Arabia (58%) [26] and previous studies from Ghana (59%, 53%) [5,16]. Additionally, we found a high rate of MDR among those with CoNS infections (52%), although this was lower than a recent report from Myanmar (70%) [27]. This calls for improving the focus on infection prevention and control (aseptic techniques, hand hygiene, disinfection, and sterilization) among health care providers and mothers [4,27].

A previous study from Ghana showed a similar proportion of MDR of 53% [5]. This high proportion of MDR, including methicillin and vancomycin resistance to *S. aureus*, makes it difficult to choose the most appropriate empiric antibiotic regimen and argues for the need for ongoing local AMR surveillance and antibiotic stewardship. These findings could guide improved antibiotic selection in the future. Fortunately, there was a much lower proportion of Gram-negative infections (11%, 15/139) in this study, which has been associated with worse neonatal outcomes [30].

## 5. Conclusions

In this study of neonatal sepsis at a large tertiary-level military hospital in Accra, Ghana, during 2017–2020, we assessed the TAT from culture request to report, culture positivity, common bacteria, and their AST along with MDR. We found the TAT to be satisfactory. Despite this, due to large numbers of neonates being discharged within three days, the culture report was not received before discharge for every second neonate. The reasons for the high rate of discharge before receiving the culture report need to be explored. One in three neonates with suspected sepsis were culture-confirmed and three in five neonates with confirmed sepsis were late-onset. The most common pathogen was CoNS, followed by *S. aureus*, of which one in two of the isolates were MDR. This indicates the need for better infection prevention and control along with the need for ongoing local AMR surveillance and antibiotic stewardship.

## Figures and Tables

**Figure 1 ijerph-19-11659-f001:**
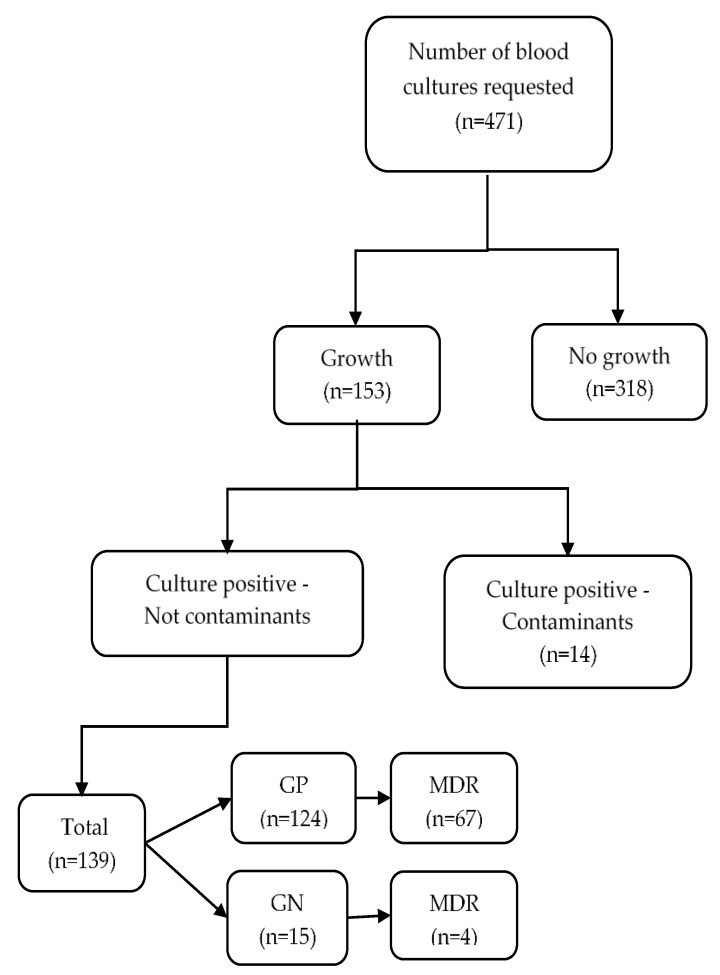
Culture positivity and multidrug resistance among neonates with suspected sepsis who underwent culture sensitivity testing at the 37 Military Hospital, Accra, Ghana (2017–2020). GP = Gram-positive isolate, GN = Gram-negative isolate, MDR = multidrug resistant (resistance to at least one antibiotic from three or more antimicrobial categories).

**Table 1 ijerph-19-11659-t001:** Baseline characteristics and exit outcomes of neonates with suspected sepsis who underwent culture and antibiotic sensitivity testing at the 37 Military Hospital, Accra, Ghana (2017–2020).

Characteristics	N	(%)
Total	471	(100.0)
Year of admission		
2017	79	(16.8)
2018	58	(12.3)
2019	220	(46.7)
2020	114	(24.2)
Age in days		
<7	337	(71.5)
7–13	81	(17.2)
14–20	17	(3.6)
21–28	36	(7.6)
*Mean (SD)*	*5.4*	*(7.0)*
Sex		
Male	248	(52.7)
Female	223	(47.3)
Birth weight in kilograms		
Very low (1.00–1.49)	63	(13.4)
Low birth weight (1.50–2.49)	85	(18.0)
Normal (≥2.50)	323	(68.6)
Mean (SD)	2.92	(1.0)
Name of ward		
NICU	198	(42.0)
PEU	268	(57.0)
POPD/ Yeboah ward/ outside	3	(0.6)
Not recorded	2	(0.4)
Type of beneficiary		
Entitled *	81	(17.2)
Non entitled	389	(82.6)
Not recorded	1	(0.2)
Category of sepsis		
Early onset (<3 days)	228	(48.4)
Late-onset (within 3–28 days)	243	(51.6)
Hospital exit outcomes		
Clinically improved and discharged	439	(93.2)
Died	32	(6.8)

NICU = neonatal intensive care unit, PEU = paediatric emergency unit, POPD = paediatric outpatient department, SD = standard deviation; * Entitled = dependents of serving and ex-military personnel as well as civilian employees.

**Table 2 ijerph-19-11659-t002:** Baseline characteristics of neonates with culture-confirmed sepsis, stratified by early and late-onset sepsis, at the 37 Military Hospital, Accra, Ghana (2017–2020).

	Total	Early Onset		Late-Onset	
Characteristics	N	N	(%)	N	(%)
Total	139	54	(38.8)	85	(61.2)
Year of admission					
2017	28	9	(32.1)	19	(67.9)
2018	12	4	(33.3)	8	(66.7)
2019	62	24	(38.7)	38	(61.3)
2020	37	17	(45.9)	20	(54.1)
Sex					
Male	77	30	(39.0)	47	(61.0)
Female	62	24	(38.7)	38	(61.3)
Birth weight in kilograms					
Very low (1.00–1.49)	22	17	(77.3)	5	(22.7)
Low birth weight (1.50–2.49)	17	5	(29.4)	12	(70.6)
Normal (≥2.50)	100	32	(32.0)	68	(68.0)
Mean (SD)					
Name of ward					
NICU	42	34	(81.0)	8	(19.0)
PEU	95	20	(21.1)	75	(78.9)
POPD/Yeboah ward/outside	1	0	(0.0)	1	(100.0)
Not recorded	1	0	(0.0)	1	(100.0)
Type of beneficiary					
Entitled *	27	10	(37.0)	17	(63.0)
Non entitled	112	44	(39.3)	68	(60.7)
Gram reactivity					
Positive	124	44	(35.5)	80	(64.5)
Negative	15	10	(66.7)	5	(33.3)

Row percentages (denominators are the values in column N); NICU = neonatal intensive care unit, PEU = paediatric emergency unit, POPD = paediatric outpatient department, SD = standard deviation; * Entitled = dependents of serving and ex-military personnel as well as civilian employees.

**Table 3 ijerph-19-11659-t003:** Antibiotic susceptibility testing patterns of Gram-positive isolates (*n* = 124) among neonates with culture-confirmed sepsis at the 37 Military Hospital, Accra, Ghana (2017–2020).

Isolates	CoNS (*n* = 68)			*S. aureus* (*n* = 45)			*Enterococcus* spp. (*n* = 7)			*S. agalactiae* (*n* = 2)			*S. mitis* (*n* = 1)			*S. faecalis* (*n* = 1)		
Antibiotics	Test	RES	(%)	Test	RES	(%)	Test	RES	(%)	Test	RES	(%)	Test	RES	(%)	Test	RES	(%)
Amoxacillin clavulanic acid	60	23	(38)	39	13	(33)	6	2	(33)	1	1	(100)	1	0	(0)	1	1	(100)
Ampicillin	68	54	(79)	45	40	(89)	7	6	(86)	2	2	(100)	1	1	(100)	1	1	(100)
Cefotaxime	61	35	(57)	44	27	(61)	6	4	(67)	1	1	(100)	1	0	(0)	1	1	(100)
Cefoxitin	-	-	-	45	13	(29)	-	-	-	-	-	-	-	-	-	-		-
Chloramphenicol	68	27	(40)	45	28	(62)	7	5	(71)	2	0	(0)	1	1	(100)	1	1	(100)
Ciprofloxacin	63	9	(14)	44	10	(23)	7	2	(29)	2	1	(50)	1	0	(0)	1	0	(0)
Cotrimoxazole	63	47	(75)	44	29	(66)	7	5	(71)	2	2	(100)	1	1	(100)	1	1	(100)
Erythromycin	62	32	(52)	41	32	(78)	6	5	(83)	1	1	(100)	1	0	(0)	1	1	(100)
Gentamicin	63	22	(35)	44	17	(39)	7	3	(43)	2	1	(50)	1	1	(100)	1	0	(0)
Levofloxacin	63	8	(13)	44	8	(18)	7	0	(0)	2	0	(0)	1	0	(0)	1	0	(0)
Oxacillin	60	28	(47)	41	24	(59)	6	3	(50)	1	1	(100)	1	1	(100)	1	1	(100)
Penicillin	60	42	(70)	41	33	(80)	6	6	(100)	1	1	(100)	1	1	(100)	1	1	(100)
Tetracycline	68	44	(65)	45	22	(49)	7	5	(71)	2	2	(100)	1	1	(100)	1	1	(100)
Vancomycin	61	27	(44)	41	31	(76)	6	4	(67)	1	1	(100)	1	1	(100)	1	1	-

Row percentages; CoNS = coagulase-negative Staphylococcus, RES =resistant, *n* = number of isolates.

**Table 4 ijerph-19-11659-t004:** Antibiotic susceptibility testing patterns of Gram-negative isolates (*n* = 15) among neonates with culture-confirmed sepsis at the 37 Military Hospital, Accra, Ghana (2017–2020).

Isolates	*K. pneumoniae* (*n* = 6)			*A. baumannii* (*n* = 3)			*E. coli* (*n* = 2)			*Aeromonas veronii bv sobria* (*n* = 1)			*Pseudomonas* spp. (*n* = 1)			*M. catarrhalis* (*n* = 1)			*Salmonella* spp. (*n* = 1)		
Antibiotics	Test	RES	(%)	Test	RES	(%)	Test	RES	(%)	Test	RES	(%)	Test	RES	(%)	Test	RES	(%)	Test	RES	(%)
Amikacin	6	0	(0)	3	0	(0)	2	0	(0)	1	0	(0)	1	0	(0)	1	0	(0)	1	0	(0)
Amoxacillin clavulanic acid	6	1	(17)	3	0	(0)	2	0	(0)	1	1	(100)	1	1	(100)	1	1	(100)	1	1	(100)
Ampicillin	6	6	(100)	3	3	(100)	2	1	(50)	1	1	(100)	1	1	(100)	1	1	(100)	1	1	(100)
Cefotaxime	5	3	(60)	3	0	(0)	2	1	(50)	1	1	(100)	1	0	(0)	1	1	(100)	1	1	(100)
Cefuroxime	6	2	(33)	3	1	(33)	2	1	(50)	1	0	(0)	1	0	(0)	1	0	(0)	1	0	(0)
Chloramphenicol	6	3	(50)	3	2	(67)	2	1	(50)	1	1	(100)	1	1	(100)	1	0	(0)	1	0	(0)
Ciprofloxacin	6	2	(33)	3	0	(0)	2	0	(0)	1	0	(0)	1	0	(0)	1	0	(0)	1	0	(0)
Cotrimoxazole	5	4	(80)	3	2	(67)	2	0	(0)	1	1	(100)	1	0	(0)	1	0	(0)	1	1	(100)
Gentamicin	6	2	(33)	3	0	(0)	2	0	(0)	1	0	(0)	1	0	(0)	1	1	(100)	1	0	(0)
Levofloxacin	6	0	(0)	3	0	(0)	2	0	(0)	1	1	(100)	1	0	(0)	1	0	(0)	1	0	(0)
Meropenem	6	0	(0)	3	0	(0	2	0	(0)	1	0	(0)	1	0	(0)	1	0	(0)	1	0	(0)
Tetracycline	5	4	(80)	3	1	(33)	2	0	(0)	1	1	(100)	1	0	(0)	1	0	(0)	1	0	(0)

Row percentages; RES = resistant, *n* = number of isolates.

**Table 5 ijerph-19-11659-t005:** Multidrug resistance among neonates with suspected sepsis who underwent culture and antibiotic sensitivity testing at the 37 Military Hospital, Accra, Ghana (2017–2020).

Bacteria Isolates	Number of Isolates	MDR Isolates	
		*n*	(%)
**Overall**	139	71	(51.1)
**Gram-positive isolates**	**124**	**67**	**(54.0)**
Coagulase Negative *Staphylococcus*	68	35	(51.5)
*Staphylococcus aureus*	45	23	(51.1)
*Enterococcus* spp.	7	5	(71.4)
*Streptococcus agalactiae*	2	2	(100.0)
*Streptococcus mitis*	1	1	(100.0)
*Streptococcus faecalis*	1	1	(100.0)
**Gram-negative isolates**	**15**	**4**	**(26.7)**
*Klebsiella pneumoniae*	6	2	(33.3)
*Acinetobacter baumannii*	3	1	(33.3)
*Escherichia coli*	2	0	(0.0)
*Aeromonas veronii bv sobria*	1	1	(100.0)
*Pseudomonas* spp.	1	0	(0.0)
*Moraxella catarrhalis*	1	0	(0.0)
*Salmonella* spp.	1	0	(0.0)

MDR = multidrug resistant, *n* = number of isolates.

**Table 6 ijerph-19-11659-t006:** Resistance of isolates to antibiotics, stratified by AWaRe category, among neonates with confirmed sepsis at the 37 Military Hospital, Accra, Ghana (2017–2020).

Classes of Antibiotics	Antibiotics	AWaRe Category	Tests	Resistant	
			N	*n*	(%)
Aminoglycosides	Amikacin	Access	15	0	(0)
	Gentamicin	Access	133	47	(35.3)
Amphenicols	Chloramphenicol	Access	139	70	(50.4
Beta-lactams—Beta lactamase inhibitor	Amoxacillin-clavulanic acid	Access	148	20	(13.5)
Carbapenems	Meropenem	Watch	15	0	(0)
Cephalosporins-2nd Generation	Cefoxitin	Watch Access	45	13	(28.9)
	Cefuroxime	Watch	15	4	(26.7)
Cephalosporins-3rd Generation	Cefotaxime	Watch	128	75	(58.6)
Fluoroquinolones	Ciprofloxacin	Watch	133	24	(18.0)
	Levofloxacin	Access	133	17	(12.8)
Glycopeptides	Vancomycin	Watch	111	65	(58.6)
Penicillin	Penicillin	Access	110	84	(76.4)
	Ampicillin	Access	137	118	(84.9)
	Oxacillin	Access	110	58	(52.7)
Macrolides	Erythromycin	Watch	112	71	(63.4)
Tetracyclines	Tetracycline	Access	138	81	(58.7)
Sulfonamides	Cotrimoxazole	Access	132	93	(70.5)

AWaRe = Access, Watch, Reserve, CoNS = coagulase negative Staphylococcus, *n* = number of isolates.

## Data Availability

The data and codebook used in this study have been shared as Appendix A.

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
