# Peer review of "Sepsis among Neonates in a Ghanaian Tertiary Military Hospital: Culture Results and Turnaround Times"

_ijerph, 2022, doi:10.3390/ijerph191811659_

Round 1
Reviewer 1 Report
Overview
Francis Kwame Morgan Tetteh, et al. performed a laboratory-records based analysis on neonatal sepsis from a single hospital in Ghana. They focused on exploring culture-confirmation, turnaround time of laboratory results, species, and drug resistance. It is refreshing to see a paper focusing on improving hospital systems, given the emphasis on turnaround time and the detail provided in the paper regarding species and drug resistance is valuable. There are a few items that the authors could clarify in their manuscript. Please find specific comments below:
Introduction
· Consider rephrasing “LOS is usually acquired through healthcare associated infections” and perhaps additional language in the first paragraph. Sepsis itself is not an infection (so cannot be acquired), but is rather a condition brought on by the body’s reaction to an infection.
· Are there citations available specifically about the rate of neonatal sepsis-related mortality? Paragraph 2 provides data for sepsis cases and for neonatal mortality, but does not connect sepsis to mortality rates like paragraph 3 suggests it does. A citation that connects the two would be helpful to add for readers.
· I cannot find the $469 billion figure or any economic cost estimates in citation #7. Was the wrong citation used here?
· It is unclear what the added benefit of including Annex 1 is, nor why Annex 1 does not align with the designations on the official WHO AWaRe website. I would recommend simply citing AWaRe as a resource but not duplicating WHO’s existing material.
Methods
· 2.2.2: Because of the focus on turnaround time of results, it would be beneficial to explicitly state whether all laboratory testing was done on-site or if any send-out testing was performed.
· 2.2.3: Could the authors clarify whether they are incubating all vials for 5 days or whether the vials are checked before then (e.g. daily) and if growth is detected, the next steps are performed at that time. Because of the emphasis on turnaround time, it is important to be clear about any laboratory practices that could effective turnaround time.
· 2.3: Is this intended to mean that all neonates must have sepsis at the time of admission to the NICU to be included? That would seem to exclude any neonates with late onset sepsis. If instead the authors mean “all admitted neonates with suspected sepsis”, I would recommend adjusting the language to clarify this.
· 2.3: Can the authors clarify if neonates had to be in the NICU in order to be eligible for this study? Section 2.2.2. focuses on the NICU, but the NICU is not explicitly mentioned in 2.3 and, in 2.4, ward is listed as a variable and it is unclear whether that was simply used for inclusion/exclusion or if neonates could have been in other units and still included in the study. If other wards could be included, it would be good to have a description of those as well.
Results
· Figure: The figure says it is Figure 37, but it is the only figure in the paper.
· Figure: I would recommend removing the n=71 MDR box as these appear to be the same isolates as those in the Gram positive and Gram negative MDR boxes.
· To clarify what the 72% is associated with, I would recommend changing “The majority (72%) of neonates were < 7 days of age, of normal birth weight (69%) and from the non-entitled group (83%)” to “The majority of neonates were < 7 days of age (72%), of normal birth weight (69%) and from the non-entitled group (83%).” Otherwise, it could be interpreted as 72% of neonates had all three of those characteristics.
· Turnaround time is stated in the introduction as one of the main items to evaluate in this paper, but there is no table or figure that shows any of those results. I would recommend exploring how to incorporate that data into a table or figure.
· 3.3: Rather than explain using chi square testing within this section, that should be included in
the methods.
· Were there differences in turnaround time, result receipt, or death across NICU v. PEU? Table 2 suggests that the patients in these wards may be very different populations.
· 3.5: Should the p-value in the 4th sentence say “p=0.121”?
· Table 3 and 4: The resistant percentages appear to be calculated in the same way for all organisms, but only one percent column has a footnote. Are the others calculated differently? If so, can that be clarified?
· Table 3 and 4: The tables may be more easily readable if drug grouping subheaders, alphabetical order, or numeric order is used for the rows.
· Table 6: If the authors are going to use different AWaRe categories than WHO (e.g. WHO lists cefoxitin as Watch while table 6 lists it as Access), then either a very clear explanation must be given about why the different categorizations are used and perhaps it should not be called the “AWaRe category” but rather something that indicates the alternative method (e.g. “AWaRe adjusted for local Ghanaian context”).
· Table 6: Oxacillin is missing its AWaRe category in the table. Please add that in.
Discussion
· 4.1 and 4.3: Remove “second” from “…the report was not received before discharge in every second neonate” and “…the culture report was not received before discharge in every second neonate.” Perhaps the authors mean “every single neonate”? This also occurs in the conclusion.
· 4.2: I would not categorize this as secondary data given the lead authors work for this hospital and had the original data available from the medical and laboratory records.
· 4.3: As three is the median, not the maximum, I would recommend rephrasing “These results indicate that a culture-confirmed neonate with sepsis may have to wait up to three days before a more specific and narrow spectrum antibiotic choice can be made.”
· One of the most striking findings for me was the four-fold increase in suspected sepsis cases from 2018 to 2019 (and five-fold increase in confirmed cases). Yet this is not mentioned in the discussion. Do the authors have thoughts on why the jump occurred? Do they believe it is artificial (e.g. lab capacity increased so more lab tests were ordered across the hospital) v. a true increase (e.g. staffing shortages meant infection control measure may have faltered)? I would highly encourage the authors to include some discussion about this as it seems important for interpreting the findings.
Annex 2
· Admdura: Multiple records say “01/01/1800.” Should this be a different value?
Reviewer 2 Report
I have read this manuscript with interest. I consider that the topic addressed is current and relevant, and in general, the manuscript is well written. Consequently, I only have 2 comments:
First, as shown in Results, section 3.2. It is noteworthy that there is apparently an association between TAT and culture positivity. In fact, TAT was three days for culture-positive samples and five days for culture negative samples.
Second, There also appears to be another association between TAT and the risk of death, because “For the 32 neonates who died, 19 received the culture report after death and 11% of those had positive cultures”.
Understandably, due to the cross-sectional design of the study, these questions are difficult to explore. However, it would be interesting if the authors hypothesized a possible explanation in this regard.
Finally, please correct this expression: “139 culure-confirnmed neonates…”
Author Response
#Response to Reviewer 2 Comments
Comment 1: First, as shown in Results, section 3.2. It is noteworthy that there is apparently an association between TAT and culture positivity. In fact, TAT was three days for culture-positive samples and five days for culture negative samples.
Response 1: We did notice that TAT was three days for culture-positive and five days for culture negative. However, this is to do with the processes and not with any association between the two. As detailed in the settings, the laboratory waits for five days before declaring culture-negative. Hence, culture negative samples are expected to have a median five day delay. If there is any growth in any of these five days, the species identification and AST is performed. Hence for culture positive it is expected to be less than five days. We hope this clarifies.
Comment 2: Second, there also appears to be another association between TAT and the risk of death, because “For the 32 neonates who died, 19 received the culture report after death and 11% of those had positive cultures”. Understandably, due to the cross-sectional design of the study, these questions are difficult to explore. However, it would be interesting if the authors hypothesized a possible explanation in this regard.
Response 2: We did consider this. But we decided against discussing this. There were other relevant (more important findings) that merit disucssion. The authorgroup thought that by inclusion of the hypothesization as suggested, may make the discussion section too complicated to comprehend (also cross-sectional nature is a major limitation). We hope this is fine
Comment 3: Finally, please correct this expression: “139 culure-confirnmed neonates…”
Response 3: We have made the correction in spelling. Thank you.
